# Including patient experiences from online sources in guidelines: A natural language processing study on scabies

Lea Lösch[1]*, Carolina J. G. Kampman[2], Elena Syurina[1], Florian A. Kunneman[3], Helena Liekens[1], Taylor Doughty[4], Mart L. Stein[2], Linda Smid[2], Aura Timen[1,2,5], Teun Zuiderent-Jerak[1]

1 Athena Institute, Faculty of Science, Vrije Universiteit Amsterdam, The Netherlands, 2 Centre for Infectious Disease Control (CIb), National Institute of Public Health and the Environment (RIVM), Bilthoven, The Netherlands, 3 Language and Communication, Department of Languages, Literature and Communication, Utrecht University, The Netherlands, 4 Department of Computer Science, Vrije Universiteit Amsterdam, The Netherlands, 5 Department of Primary and Community Care, Radboud University Medical Centre, Nijmegen, Netherlands

* lea.loesch@vu.nl

## Abstract

### Objective

Including patients' experience-based knowledge in the development of clinical and public health guidelines has been shown to enhance the quality, relevance, and applicability of guidelines. However, the meaningful and methodologically sound inclusion of patient experiences remains a challenge. This study aimed to showcase the potential of NLP methods as an innovative tool for guideline development to gain insights into patients' experiential knowledge and to incorporate this into the guideline development process.

### Methods

For the revision of the Dutch public health guideline for scabies, we analyzed patients' experiences with scabies infestation shared on "dokter.nl", the Netherlands' largest online health community, between December 4, 2014, and May 19, 2023. Structural topic modelling was performed to discern thematic clusters from these patient experiences.

### Results

We obtained 5781 unique posts on scabies and identified 13 major themes raised in forum conversations. The most prevalent themes revolved around community support (11.2%), uncertainty about treatment plans (11.1%) and coping with itching (11%). Recognizing scabies, alternative remedies, and decontamination measures were also issues frequently raised. The analysis highlighted the burden of disease

**Data availability statement:** The data underlying this study are available from the following repository: https://doi.org/10.6084/m9.figshare.25722240.v1.

**Funding:** This research was funded by ZonMw, The Netherlands Organization for Health Research and Development, under grant agreement number 516022526. The funders had no role in study design, data collection and analysis, decision to publish, or preparation of the manuscript.

**Competing interests:** The authors have declared that no competing interests exist.

and treatment—particularly the psychosocial burden—associated with scabies. This offered guideline developers an unprecedented insight into patients' experiences resulting in alterations to the Dutch public health guideline for scabies.

## Conclusion

Previous studies have highlighted the benefits of the integration of experiential knowledge for guideline development. Our study provides a novel method to make this type of knowledge accessible and usable for medical guideline development, without additionally burdening patients.

## Introduction

Clinical practice and public health guidelines aim to provide recommendations based on the best available evidence to support health professionals in (clinical) decision-making and ultimately improve patient care and public health. Guideline developers thereby draw primarily on evidence from clinical research, such as randomized controlled trials and systematic reviews thereof, and to a lesser extent on cohort studies, case series and expert opinions [1,2]. Despite the value of evidence from clinical research, there is increasing recognition of the importance of incorporating the experiences and values of relevant stakeholders, such as patients and healthcare professionals, alongside clinical evidence [3]. Patients can, for instance, offer insights into the burden of disease and treatment, areas of concern, needs, preferences, and patient-relevant outcomes as well as prioritize guideline topics. Guidelines that integrate this type of knowledge have been shown to be of higher quality, utility, and relevance to medical practice [4,5]. As a result, significant efforts have been made to develop methods and tools for more effectively incorporating experiential knowledge from diverse actors. Current strategies for involving the public and patients at various stages of guideline development include consultative methods (such as interviews, focus groups, questionnaires and comments on drafts), participatory methods (e.g., involving patient representatives as panelists in the guideline development process) and communication strategies (providing information to patients to support their individual healthcare decisions), or a combination of these [6]. Moreover, insights into experiential knowledge can also be gained through reviewing relevant literature and synthesizing qualitative evidence [7]. The degree and type of involvement may depend on the topic of the guideline, the availability of patient organizations, time, and resources. Despite the recognized added value, incorporating patient experience and preferences remains a major challenge in the development of clinical practice and public health guidelines, hindered by constraints such as a lack of funding and expertise, methodological unclarity, uncertainty about roles and expectations, and the complexities of integrating this knowledge with other forms of evidence in prevailing evidence appraisal frameworks [8,9].

The challenge of effective and meaningful patient engagement is exacerbated in the case of developing guidance for infectious diseases. Patient organizations are

uncommon, making it harder to recruit and involve patients, which is often needed for traditional engagement methods. In some cases, a wider scope of involvement is needed, e.g., of contacts of an infected case or the general public (e.g., for a COVID-19 guideline), which is more time-consuming and complex to realize. Lastly, when developing guidance amid constantly evolving evidence coupled with the urgency to provide rapid guidance during outbreaks like COVID-19 or Mpox, traditional engagement methods have proven difficult to implement in a short timeframe [10,11].

This study presents an innovative approach to address these challenges of including patients' experiential knowledge in guideline development when traditional engagement methods are challenging to implement. This approach uses AI-based methods, particularly from the field of natural language processing (NLP) and was applied and examined in the case of the revision of the Dutch public health guideline for scabies. Over the past decade, numerous studies have employed NLP techniques to analyze patient experiences of, e.g., healthcare services [12], health technologies [13], and diseases [14] from patient-generated text data in surveys, on social media and online health communities. Despite substantial research and discussions on the potential of these AI-based methods, there persists a well-known implementation gap [15]. Many of these techniques offer marginal or unclear practical utility and few studies test medical NLP applications in real-world settings, leaving results and discussions largely confined to academic discourse [16].

In the field of guideline development, the AI technologies currently being developed primarily focus on supporting the collection and synthesis of scientific literature [17,18]. More recently, there has been an emerging interest in utilizing "real-world evidence" for guideline development, which, however, is primarily targeted at quantitative, routinely collected data from clinical registries and less at the inclusion of patient experiences [19]. There is no literature available that explores the use of AI-based methods to integrate patients' experiential knowledge into guideline development.

In this study, we demonstrate the implementation and added value of employing NLP methods to learn about patients' experiences in the case of the revision of the Dutch public health guideline for scabies. Scabies is a skin infestation, caused by the human scabies mite (*Sarcoptes scabiei var. hominis*). In common scabies, the scabies mites are transmissible through skin-to-skin contact. Because the mite can survive outside the human body, medical treatment of the patient and contacts is necessary, in addition to decontamination measures such as machine washing of clothing and cleaning contact surfaces [20]. Scabies treatment can be a challenging and lengthy process, involving prolonged efforts to eliminate scabies that significantly affect the individual and their surroundings. In recent years, the incidence of scabies has risen continuously worldwide as well as in many European countries, including the Netherlands [21]. The reported figures are likely still underestimated due to insufficient monitoring and data. Unfortunately, there are no patient organizations for scabies in the Netherlands, and furthermore, the disease carries a social stigma that may inhibit willingness to participate in available patient engagement methods. Our method of systematically gathering large amounts of diverse patient experiences shared online holds great potential in this context and may overcome potential limitations of traditional patient engagement methods.

The aim of this study is to showcase the potential of NLP methods as an innovative, additional tool for guideline development to gain insights into patients' experiential knowledge and preferences and to incorporate these into public health guidance, by presenting the results and practical application of these techniques in the context of the revision of the Dutch public health guideline on scabies.

## Methods

### Study design and data source

For the revision of the Dutch public health guideline for scabies in the Netherlands, we sought to understand the experiences of patients infected with scabies and their close contacts by analyzing their reports shared online using NLP techniques. The methodology adopted in this retrospective NLP study is structured into the following phases: 1) preliminary mapping and scanning of online sources, 2) data retrieval and preparation for analysis, 3) implementation of unsupervised

machine learning to discern thematic patterns [22], and 4) incorporating the findings into the guideline development process. Throughout this study, we adopt a human-in-the-loop approach, where the researchers play an active role at various stages of the NLP workflow. Automated analysis constantly alternates with the researchers' review, analysis and contextualization of the data, a common approach in NLP research to enhance the accuracy, quality, and relevance of the analyses [23].

After a broad exploration of various channels such as YouTube, Instagram, Twitter, Reddit, LinkedIn, Facebook, Dutch news websites such as "nu.nl", and online forums, the public forum "dokter.nl" (translating to "doctor.nl", referring to a medical doctor) was selected as a promising open data source. While scabies was not or only occasionally discussed on the other channels, dokter.nl contained numerous comments sharing experiences. Dokter.nl is the Netherlands' largest online health community where all sorts of medical issues are discussed, with more than 1 000 000 visitors and 600 000 new forum posts per month. It is a public, low-threshold forum that allows posting anonymously under a chosen username and lent itself well to this study as the comments were extensive, diverse, centered around the topic of scabies infection, and accessible due to the public status of the forum. The forum administrators were briefed on this research and data was collected and used in compliance with the Terms and Conditions set forth by Dokter.nl. All data from the sub-thread dedicated to scabies, since the initiation December 4, 2014, to May 19, 2023, were retrieved using Python (version 3.10) and the *BeautifulSoup* library [24]. This resulted in unique 5781 forum entries, 96.16% of which were posted since 2020. We retained the post text, the date, and the link to the original post; the usernames have been omitted to enhance anonymity.

## Data analysis

The retrieved text data underwent several common pre-processing steps using the *quanteda* R package [25] to prepare them for analysis: duplicate texts were removed, only Dutch-language text was retained, words were lowercased, numbers, punctuation, URLs, and symbols were removed, and words were reduced to their word stems. In addition, a list of so-called "stop words", i.e., frequently used words such as articles or prepositions that carry no significant meaning, has been removed. Finally, words occurring in fewer than 0.35% or over 99% of the documents were excluded, as they are not indicative of patterns and themes in the data. This yielded a final dataset comprising 5644 unique records.

Subsequently, we performed topic modelling to analyze the content of people's experiences related to scabies. We ran and tested two types of topic models, Latent Dirichlet Allocation (LDA) [26] and Structural Topic Modelling (STM) [27]. The resulting topics were analyzed and compared by two of the authors, indicating that STM yielded more interpretable results—meaning the topics were more coherent, distinct from one another, and easier to meaningfully label. STM is a type of probabilistic topic modelling using unsupervised machine learning to discover thematic clusters in a large collection of documents (in our case, forum posts). In contrast to other topic modelling algorithms such as Latent Dirichlet Allocation, STM can accommodate corpus structure by incorporating document-specific metadata (e.g., publication date) into the model estimation process. This can aid the prediction process [28] and also provides the option to subsequently estimate the effects of those covariates on, e.g., topic prevalence. The thematic clusters resulting from the topic modelling, or "topics", are formed based on the co-occurrence of words. This rests on the linguistic assumption that words that frequently appear together across a set of documents are likely to be related in meaning. Topic modelling is an exploratory approach that is particularly suitable for gaining insights into inherent patterns and themes from large volumes of unstructured text data.

Prior to conducting topic modelling, the number of topics ($K$) to be modelled must be specified. Following Roberts et al. (2019) we selected the number of topics that maximizes the measures of semantic coherence—indicating how frequently the most probable words of a topic co-occur—and exclusivity—the extent to which words are exclusive to a particular topic. The highest values for both measures were achieved at $K = 13$. The structural topic modelling was run with the *stm* package in R (version 4.3.0) [27].

To understand the resulting topics and check the model's validity, we manually inspected 20 documents most strongly associated with a topic for each of the 13 topics. Additionally, we examined the 10 most probable terms and the 10 most

frequent but exclusive terms per topic in the context of the text corpus. Based on the analysis of the top terms in conjunction with the content of the representative documents, descriptive labels reflecting the underlying themes were assigned to each topic.

### Ethics approval

This research underwent an ethical self-assessment provided by the Ethics Review Committee of the Faculty of Science (BETHCIE) at Vrije Universiteit Amsterdam. The self-assessment did not raise any concerns, confirming that no further formal review by BETHCIE was required. This determination was based on the fact that data were exclusively obtained from a publicly accessible data source and because no personal and identifiable data (e.g., names, usernames, IP addresses) were collected. Consequently, formal informed consent was not required nor collected. To further safeguard users' privacy, we do not publish specific comments that could be used to identify the original user and only share comment IDs in our data set.

### Results

Using structural topic modelling, we identified 13 overarching topics that were raised in the online forum on scabies (Table 1). In the following, we will focus on those topics that were most pertinent to the revision of the public health guideline for scabies, as they led to specific changes in the guideline text.

Many of the topics identified revolve around and point towards the various aspects of the psychosocial burden experienced by people infected with scabies. In topic 1, with the highest topic proportion (11.2%), forum users express their

**Table 1. Topics identified in patients' experiences shared on dokter.nl, their assigned labels, proportions, and the most frequent and simultaneously most exclusive (FREX) words, translated from Dutch to English (n = 5644).**

| Topic and label | Corpus, n (%) | FREX terms |
|---|---|---|
| 1 Community support | 632 (11.2%) | thanks, fine, say, read, hopefully, indeed, sounds, hope, sucks, haha [*written expression of laughing*] |
| 2 Combination treatment plans | 627 (11.1%) | pills, ivermectin, second, tablets, permethrin, smear, times, cream, 2nd, cure |
| 3 Itching | 621 (11%) | itching, worse, friend, awake, bother, night, daytime, scratching, night, sleep |
| 4 Recognizing scabies | 553 (9.8%) | my, red, itchy, spot, small, bump, legs, belly, back, photo |
| 5 Availability of information/ drug dosages | 491 (8.7%) | one, resistance, information, leaflet, most, site, understand, article, study, people |
| 6 Old vs active infection | 474 (8.4%) | dot, black, sees, corridor, see, clear, out, dead, egg, suspicious, sure |
| 7 Self-made creams | 395 (7%) | oil, sulfur, tea, tto, tree, coconut oil, sulfur powder, vera, smear, cream |
| 8 Decontamination measures | 378 (6.7%) | bags, washed, bedding, clothing, sofa, degrees, wash, clean, pillow, clothes |
| 9 Infected children | 367 (6.5%) | children, hospital, literally, germany, youngest, clean, family, absolutely, permethrin, 1st |
| 10 Medicines for treatment | 367 (6.5%) | perm, bb, better, permethrin, inflammation, questions, environment, accidental, treat, time |
| 11 Four-Member Family | 299 (5.3%) | permethrin, husband, family, got, half, son, kid, year, corticosteroids, 2021 |
| 12 Follow-up skin problems | 265 (4.7%) | dry, sensitive, fungus, feet, eczema, infection, spots, skin, allergic, pimples |
| 13 Reinfection & other | 175 (3.1%) | symptoms, contact, hard, wait, sign, does, physical, want, topic, look |

appreciation for the forum community as a space for exchanging experiences, receiving support, consolation, understanding, and encouragement. Severe itching—one of the most common symptoms of scabies infection—is the third largest topic (11%) in our analysis. Those affected describe feeling dirty and that the itching usually worsens at night, which can lead to sleep deprivation, as reported by several forum users (topic 3). Patients and their close contacts describe a constant state of alarm and anxiety caused by uncertainty about the success of the treatment and the possible return of the mite (topic 6). The fear of (re)infecting others or themselves combined with feeling ashamed and dirty has led some patients to self-isolate, sometimes for weeks or months. They express that unsuccessful treatment(s) led to despair, frustration, and exhaustion (especially for members of large families) from trying to closely adhere to treatment plans (topic 2) and decontamination measures (topic 8). In some cases, this fear and desperation drove individuals to exceed recommended treatment frequency and doses hoping to enhance treatment effectiveness (topic 5), and to experiment with alternative remedies.

The exploration of alternative substances in the form of homemade creams surfaced as a distinct theme in our analysis (topic 7). Among the most frequently mentioned substances were sulfur (seen in 251 [63.5%] of the posts assigned to this topic), tea tree oil (in 239 [60.5%] posts), and neem oil (in 62 [15.7%] posts). Patients resorted to these remedies when medication was unavailable, e.g., when travelling abroad, or as an additional treatment method.

Lastly, forum users reported very sensitive and damaged skin following scabies medication and treatment (topic 12, with a topic proportion of 4.7%). Despite successful treatment, those affected still had to deal with considerable skin problems, such as susceptibility to bacterial and mold infections, eczema, and allergic reactions.

## Implementation of the findings in the scabies guideline

These findings were presented to the guideline development group, which resulted in additions and alterations to the Dutch public health guideline for scabies. Since this was the first time that guideline developers obtained insight into experiential knowledge in this way, there were no standardized and structured procedures for integrating this knowledge into the guideline. Following the scoping phase, where current issues and needs are identified, guideline developers would typically consult with relevant stakeholders (through face-to-face meetings or other consultation methods) to determine and prioritize the scope and key questions of the guideline, including patient-relevant outcomes [29]. In this case, there were no processes in place for dialogue and consultation with the forum users, who had provided the experiential knowledge, deeming their engagement not feasible within the guideline development process and timeline (a limitation we return to in the discussion). Hence, study findings were discussed among the researchers, guideline developers, and professionals working in the field of infectious disease control on the topic of scabies. The study findings were assessed for relevance and accuracy by these professionals. Topics deemed relevant and accurate were then incorporated into the guidance.

First, the social stigmatization and psychosocial impact of scabies were recognized as profoundly affecting patients and their treatment and were therefore addressed in a separate paragraph written by a specialized dermatologist. This paragraph raises awareness of the psychosocial impact on patients' quality of life, well-being, and the course of the disease/treatment, and the link between scabies infestation and severe psychological conditions like insomnia, anxiety, and depression. Second, the guideline was enriched by providing clear advice on combatting the itching that accompanies a scabies infestation. This included skin care recommendations to provide additional relief during and after the treatment. This section was also deemed important in light of the risk of exceeding recommended doses (topic 5), due to stigma, burden of disease, and burden of treatment duration. Third, to manage the schedules for treatment and decontamination measures, a step-by-step plan for patients, both in Dutch and in English, was developed [30].

While study findings indicated that patients use alternative substances, such as tea tree oil, this was not incorporated into the guideline. Even though a recent systematic review demonstrated the promising efficacy of tea tree oil against scabies mites in vitro [31], the guideline focuses on first-choice treatments (permethrin, ivermectin) with broad scientific evidence only.

## Discussion

This is the first study to use AI-based methods to gain insights into patients' experiences to inform and contribute to the development of clinical and public health guidelines. We have demonstrated how employing NLP techniques to analyze patients' experiential knowledge online can serve as an additional, innovative method in the toolkit of methods for integrating patient knowledge into guideline development. In the case of scabies, we have showcased how the application of this method has generated valuable insights that could not have been easily obtained otherwise. These patient experiences have led to meaningful and practical additions in the guideline. The methods presented can thus contribute to meeting the persistent challenge in guideline development to incorporate experiential knowledge into guideline development in a methodologically sound manner.

A key advantage offered by this analysis of patient-generated text collected from web-based platforms is that patients describe their experiences in "natural language", i.e., accounts are spontaneously expressed in an environment that is often familiar to them rather than solicited in a research setting [32]. Patients may benefit by being able to express their experiences more candidly and anonymously in such an environment. Consequently, this unfiltered and unprompted analysis allows to uncover authentic themes that truly matter to patients. Another advantage is that this approach enables insights into patient experiences on a larger scale. The experiences of a larger and likely more diverse group of patients can be consulted, including views not represented by patient organizations. Such analyses can be carried out fairly quickly, remotely, with relatively little organizational and logistical effort, without any burden for the patients, and at low cost. While this study focuses on scabies in the Netherlands, the methods presented can be applied to a variety of other diseases, online platforms—including social media—and different national or cultural contexts. Research across countries and disease areas shows that patients increasingly use social media and online health communities to share their personal health experiences [33]. These methods may be particularly valuable in contexts and disease areas where conventional patient involvement methods prove difficult to implement. This is for example evident when patient organizations are scarce (e.g., for most infectious diseases) or when the disease is associated with a societal stigma (e.g., sexually transmitted diseases) that may hinder active patient involvement and when experiential knowledge is evolving as is the case with outbreaks of infectious diseases. Furthermore, NLP analyses can also provide more nuanced and diverse insights, precisely when patient organizations are highly organized, have strong viewpoints and advocate for specific subjects. Lastly, patient-generated online content could also be analyzed across countries—for instance, through the use of multilingual topic models [34]—but requires careful consideration of language, platform norms, and local healthcare contexts. For all such uses, a shared requirement is that there needs to be a critical number of patients discussing and sharing their experiences in online health communities, patient forums, or on social media for this method to be beneficial.

Careful awareness is needed that, while there are invaluable experiences shared online, there may be individuals—especially in cases with greater public attention—who do not share their experiences truthfully or deliberately spread false information, possibly aided by bots [35]. This can be partially mitigated by implementing bot detection filters during the data scraping process, removing duplicate posts, analyzing topics at an aggregated level giving less weight to individual posts, and by following a human-in-the-loop approach, where automated analyses constantly alternate with close reading and qualitative analysis of the data (for an extensive review of detecting and mitigating online misinformation see [36]). For example, if a post appears suspicious, researchers can revisit the forum to examine the user's profile and posting history, identifying potential red flags such as a newly created account, continuous 24/7 activity, or repetitive posting of identical content. The "human-in-the-loop" aspect is a defining strength of our method. Yet it might also pose a limitation in that it is not a fully automated process but necessitates foundational computer science expertise to ensure the safe and effective execution of these methods. Analyzing digital patient knowledge offers insights into the experiences of a larger and likely more diverse group of patients. Nevertheless, it is crucial to acknowledge that this data represents but a specific selection of patients sharing their experiences online, which also vary by channel. Despite increasing social media uptake across demographics, a selection bias towards native language writers, the

 

young, wealthy, and technologically savvy still exists [31]. Additionally, it is important to recognize that, while patients turn to online platforms for a range of reasons [33], the content shared may disproportionately highlight negative and extreme experiences, thereby overshadowing more typical cases [37,38]. Guideline developers need to be mindful of which patient groups and which experiences they may not be reaching and consider other methods from the toolkit to include these patients. Finally, it is important to recognize that, although our method can render patients' experiential knowledge and preferences more readily available, the validation and the meaningful integration of this type of knowledge into guideline development remains a challenge. One promising avenue might be to collaborate with a patient representative to review experiential knowledge from online sources and ensure that it is accurately represented and meaningfully integrated into the guideline development process. Where dedicated patient organizations do not exist, findings could be compared with analyses of additional online platforms, published literature on patient experiences, or other methods such as surveys. Innovative digital approaches could also be considered—for example, engaging directly with patients online to gather feedback or ask targeted questions.

## Conclusion

Through the purposeful application of established NLP methods, we provide guideline developers with a new, methodological sound way to incorporate experience-based knowledge into the guideline development process. We are hereby addressing the persistent challenge in evidence-based guideline development where, despite the long-recognized and proven value of this type of knowledge, its meaningful and effective integration into the guideline process remains difficult to achieve. While not a panacea, this novel approach represents a significant stride in reconciling evidence-based guideline development with its founding principles of equally drawing on clinical evidence, patient experience and preferences, and clinical expertise in the development of guidance [39], which is crucial for developing high-quality, robust, relevant, and inclusive guidelines for complex clinical and public health questions.

## Author contributions

**Conceptualization:** Lea Lösch, Elena Syurina, Florian A. Kunneman, Mart L. Stein, Aura Timen, Teun Zuiderent-Jerak.

**Data curation:** Lea Lösch, Helena Liekens, Taylor Doughty.

**Formal analysis:** Lea Lösch, Helena Liekens, Taylor Doughty.

**Funding acquisition:** Aura Timen, Teun Zuiderent-Jerak.

**Investigation:** Lea Lösch, Carolina J. G. Kampman, Elena Syurina, Florian A. Kunneman, Teun Zuiderent-Jerak.

**Supervision:** Elena Syurina, Florian A. Kunneman, Teun Zuiderent-Jerak.

**Writing – original draft:** Lea Lösch, Carolina J. G. Kampman, Linda Smid.

**Writing – review & editing:** Elena Syurina, Florian A. Kunneman, Mart L. Stein, Aura Timen, Teun Zuiderent-Jerak.

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
