## [Decision Letter · Decision Letter 0]

30 Jun 2025

Dear Dr. Lösch,

Thank you for submitting your manuscript to PLOS ONE. After careful consideration, we feel that it has merit but does not fully meet PLOS ONE’s publication criteria as it currently stands. Therefore, we invite you to submit a revised version of the manuscript that addresses the points raised during the review process.

We look forward to receiving your revised manuscript.

Kind regards,

Abayeneh Girma

Academic Editor

PLOS ONE

Journal Requirements:

2. In your Methods section, please include additional information about your dataset and ensure that you have included a statement specifying whether the collection and analysis method complied with the terms and conditions for the source of the data.

4. Please note that your Data Availability Statement is currently missing the repository name. If your manuscript is accepted for publication, you will be asked to provide these details on a very short timeline. We therefore suggest that you provide this information now, though we will not hold up the peer review process if you are unable.

Reviewers' comments:

Reviewer's Responses to Questions

**Comments to the Author**

1. Is the manuscript technically sound, and do the data support the conclusions?

Reviewer #1: Yes

Reviewer #2: Yes

2. Has the statistical analysis been performed appropriately and rigorously?

Reviewer #1: N/A

Reviewer #2: Yes

3. Have the authors made all data underlying the findings in their manuscript fully available?

Reviewer #1: Yes

Reviewer #2: Yes

4. Is the manuscript presented in an intelligible fashion and written in standard English?

Reviewer #1: Yes

Reviewer #2: No

Reviewer #1: Dear authors, thank you for sharing this work. Integrating an online forum on the topic of interest in order to get information about consumer’s view seems to be a feasible alternative. Using a natural language processing (NLP) seems plausible, although raising some questions about selection bias. Adding some references about this would help to get information about that. Generally, it would be expected that people with very bad experiences or people being very satisfied with a certain intervention would be overrepresented. Are the publications regarding this kind of bias and how to deal with it using internet platforms on medical topics? Would be helpful to add this.

255 ff: “the study findings were assessed for relevance and accuracy by these professionals. “ � were there any criteria?

Discussion: you carefully ventilating possible pitfalls and dangers of using public forums like you did. You underline that a human should assess each AI step. How could you identify fake input in dokter.NL ?

Adding some aspects mentioned above may help this work to be even more practice relevant.

Reviewer #2: General comments

This manuscript presents a novel and timely application of natural language processing (NLP) to analyze patient experiences shared online for the purpose of informing clinical guideline development. The study is well-designed, methodologically sound, and addresses a significant gap in the integration of patient experiential knowledge into guidelines, particularly for conditions like scabies where traditional patient engagement methods are challenging. The findings are impactful, leading to tangible changes in the Dutch public health guideline for scabies.

1. Generalizability: While the method is promising, the study focuses solely on scabies and a single Dutch forum. The authors should briefly discuss the potential applicability to other diseases or cultural contexts.

2. Patient engagement: The lack of direct engagement with forum users (e.g., validation of themes) is noted as a limitation. A brief discussion on how to mitigate this in future studies (e.g., involving patient advocates) would strengthen the manuscript.

3. Technical details: The preprocessing steps (e.g., stop-word removal, stemming) are described, but additional details on model validation (e.g., inter-rater agreement for topic labeling) would enhance reproducibility.

**Do you want your identity to be public for this peer review?** For information about this choice, including consent withdrawal, please see our Privacy Policy

Reviewer #1: No

Reviewer #2: No

---

## [Author Response · Author response to Decision Letter 1]

26 Sep 2025

Dear Editor,

Thank you for considering our manuscript and for the opportunity to resubmit a revised version of this article to PLOS ONE. We appreciate that both reviewers value the manuscript and consider it a good fit for PLOS ONE.

We would also like to thank the reviewers for providing valuable comments and feedback. We were able to incorporate most of the suggestions and believe that these have significantly improved the quality of our manuscript. As requested, we have uploaded a clean revised manuscript to the editorial management system, as well as a document that highlights the tracked changes made to the original submission.

Please also find a point-by-point response to all comments provided by the reviewers uploaded in the editorial management system.

Should you have any further questions or require additional clarification, please do not hesitate to contact us. We hope that our changes to the manuscript reflect the suggestions well and thank you for considering this resubmission.

Sincerely, on behalf of all authors,

Lea Lösch

---

## [Editor Report · Decision Letter 1]

8 Dec 2025

Including patient experiences from online sources in guidelines: A natural language processing study on scabies

PONE-D-25-04246R1

Dear Dr. Lösch,

We’re pleased to inform you that your manuscript has been judged scientifically suitable for publication and will be formally accepted for publication once it meets all outstanding technical requirements.

Kind regards,

Abayeneh Girma

Academic Editor

PLOS One

Additional Editor Comments (optional):

Accept for publication.

This manuscript represents a valuable contribution to interdisciplinary research at the intersection of digital health, NLP, and evidence-based guideline development. It is innovative, ethically conducted, well-executed, and clearly communicated. The study advances both methodology and practice, offering a replicable model for future guideline developers seeking to incorporate patient voices in a scalable, low-burden manner.

Reviewers' comments:

None

---

## [Editor Report · Acceptance letter]

PONE-D-25-04246R1

PLOS One

Dear Dr. Lösch,

I'm pleased to inform you that your manuscript has been deemed suitable for publication in PLOS One. Congratulations! Your manuscript is now being handed over to our production team.

Kind regards,

on behalf of

Dr. Abayeneh Girma

Academic Editor

PLOS One